# ZnS Quantum Dots Decorated on One-Dimensional Scaffold of MWCNT/PANI Conducting Nanocomposite as an Anode for Enzymatic Biofuel Cell

**DOI:** 10.3390/polym14071321

**Published:** 2022-03-24

**Authors:** Tariq Altalhi, Amine Mezni, Mohammed A. Amin, Moamen S. Refat, Adil A. Gobouri, Nimra Shakeel, Mohd Imran Ahamed

**Affiliations:** 1Department of Chemistry, College of Science, Taif University, P.O. Box 11099, Taif 21944, Saudi Arabia; ta.altalhi@tu.edu.sa (T.A.); aminemezni@yahoo.fr (A.M.); mohamed@tu.edu.sa (M.A.A.); moamen@tu.edu.sa (M.S.R.); a.gobouri@tu.edu.sa (A.A.G.); 2Department of Chemistry, Faculty of Science, Aligarh Muslim University, Aligarh 202002, Uttar Pradesh, India; gc2587@myamu.ac.in; 3Advanced Functional Materials Laboratory, Department of Applied Chemistry, Zakir Husain College of Engineering and Technology, Faculty of Engineering and Technology, Aligarh Muslim University, Aligarh 202002, Uttar Pradesh, India

**Keywords:** silver nanowires, MWCNTs, ZnS quantum dots, green route, EBFC, polyaniline/PANI, in situ polymerization, nanofiller and electrochemical behaviors

## Abstract

This study aims to design a new nanocomposite as a supporting material for wiring the enzyme to develop a bioanode in the enzymatic biofuel cell (EBFC). In this work, polyaniline-based nanocomposite was synthesized by in situ polymerization of aniline monomer. The zeta potential study of the nanofillers was carried out, which reveals the interaction between the nanofillers. The synthesized nanocomposite (MWCNT/ZnS/AgNWs/PANI) was characterized by analytical techniques, such as Fourier transform infrared spectroscopy (FTIR) and X-ray diffraction spectroscopy (XRD). Furthermore, the surface morphology and the in-depth information of the synthesized nanocomposite were displayed by scanning electron microscopy (SEM) and transmission electron microscopy (TEM), respectively. In addition, the as-synthesized nanocomposite and the designed bioanode underwent the electrochemical assessment using different electrochemical techniques such as cyclic voltammetry (CV), electrochemical impedance spectroscopy (EIS), and linear sweep voltammetry (LSV) for evaluating the electrochemical behavior of the fabricated anodes. The electrochemically regulated bioanode (MWCNT/ZnS/AgNWs/PANI/Frt/GOx) obtained an open-circuit voltage of 0.55 V and produced a maximal current density of 7.6 mA cm^−2^ at a glucose concentration of 50 mM prepared in phosphate buffer solution (PBS) (pH 7.0) as a supporting electrolyte at a scan rate of 100 mV s^−1^.

## 1. Introduction

Enzymatic biofuel cell (EBFC) is one of the advanced energy generation technologies that produce electrical energy from biofuels (substrate), namely, glucose [1], fructose [2], methanol [3], lactate [4], etc. EBFC is the subdivision of biofuel cells that employs specific enzymes as a biocatalyst for catalyzing the specific biofuel. The electrical energy produced via electrochemical catalysis in a biofuel cell is envisaged as an alternative energy source in powering various bioelectronic gadgets, for instance, artificial organs, sensors, transmitters, along with implantable medical devices and portable electronics [5,6,7]. Interestingly, glucose-based EBFC consumes glucose as fuel and converts it into gluconolactone and two electrons are generated. The electrochemical pathway of biofuel cells is outlined in Figure 1.

Glucose is cheap, renewable, and non-toxic, and more notably, it is ubiquitous, especially in physiological fluids, which makes glucose a recognizable fuel and EBFC as a power source for powering implantable devices. Moreover, EBFC works at favorable operating conditions such as ambient temperature and close to neutral pH along with no harmful emissions compared to others energy generation devices. In addition, for most electrochemical generators, a membrane/separator is typically used as a barrier layer between the cathode and anode [8]. However, for EBFC, the lack of need for a membrane simplifies the cell design to be small and compact, given that hydrogenase does not react with oxygen (an inhibitor) and the cathode enzymes (typically laccase) do not react with the fuel. This is another unique and significant feature of EBFC that should be explained to the readers. These parameters make it feasible for powering implantable devices [9].

To date, various groups of researchers have shown deep interest in this field. Mano et al., 2003 have designed an EBFC using GOx and laccase enzyme on the respective anode and cathode utilizing an osmium mediator, which provides a stable but very less power output of 0.47 μWcm^−2^ [10]. Currently, EBFCs was explored in monitoring the blood glucose levels [11]. Further, one such attempt, generating electrical energy in contact lenses, was achieved by one group of reserchers that utilizing the ascorbate and oxygen naturally available in human lachrymal liquid (tears) as fuel and oxidant [12]. Southcott et al. reported an implantable biofuel cell operating under conditions mimicking the human blood circulatory system that powered a pacemaker [13]. With the emergence of these ideas, substantial investigations have been conducted in this research area targeting the realization of glucose-based EBFC as a power source for medical implants such as insulin pumps [14], middle ear hearing devices [15], etc.

However, it is very difficult to achieve the effective electron transfer with glucose oxidase enzyme because its redox-active unit (flavin adenine dinucleotide) is surrounded by a thick protein shell that makes the direct electron transfer a challenging task; therefore, mediated electron transfer comes into play for the proper electron shuttling of FAD-dependent enzymes such as GOx. In this strategy, a redox-active substance is used whose redox potential lies in the vicinity of GOx [16]. Osmium and ferrocene are commonly used mediators with glucose oxidizing enzymes [17,18]. These redox-active mediators have some drawbacks such as leakage (leaching), incompatibility with enzymes, non-biocompatible, and high cost. To solve these issues, ferritin has been utilized as a redox-active biocompatible and inexpensive mediator. Another advantage of ferritin is that it has redox potential close to the GOx enzyme that permits it to work in the potential range of GOx and minimize the overpotential losses during operation. It is found in all cells where its main function is holding iron up to 4500 molecules via a redox mechanism. Thus, ferritin can serve as an effective electron relay between the redox center of the enzyme and the current collector, to facilitate the electron shuttling by reducing the path length [19].

Additionally, the short life span of the enzyme makes the system less durable, which arises due to the poor wiring of the enzyme to the electrode surface; therefore, this issue needs to be resolved to bring EBFCs on a commercial scale from a lab scale. Interestingly, the in-depth knowledge of nanoscience and nanotechnology and their products, i.e., nanomaterials and nanocomposites, come up as a problem solver owing to their unique inherent properties, e.g., high surface to volume ratio, high conductivity, and mechanical strength, unlike that of macroscopic/microscopic materials. Carbon-based nanomaterials such as carbon nanotubes (single and multiple walled) and graphene have shown great potential in diverse applications, for instance, ballistic protection, microwave absorbers, electrostatic charge dissipation in a space environment, fire retardation, corrosion protection, sensors, actuators supercapacitors, and biofuel cells [20]. Further, over the past few decades, metal nanoparticles such as Pt, Au, and Ag alone and in combination with other materials (nanocomposites) have been exploited immensely, which improve the resulting properties of the host materials due to their high surface to volume ratio and inherently conductive nature. Apart from metal NPs, nanostructured metal sulfides have also gained immense attention due to their appreciable pseudocapacitive performance [21]. The possibility of variable valence oxidation states in sulfides and a high theoretical capacity of sulfur compounds offer excellent capacitance behavior. Because of these properties, sulfide nanomaterials such as NiS [22], Co_3_S_4_ [23], CuS [24], and SnS [25] have been used as electrode materials for supercapacitor applications. Notably, the bandgap of these semiconductor nanoparticles is highly influenced by their defective surface; therefore, controlling the particle size of nanomaterials, their bandgap can be tuned easily to make them applicable in various applications, including energy storage devices. Recently, ZnS quantum dots undoped and doped with noble metals have been used in various applications [26]. Pathak Dinesh et al. reported silver and gold doped ZnS quantum dots were used for photovoltaic applications [27,28]. Zhang and coworkers used robust tin oxide nanoparticles dispersed on anode materials for sodium–ion batteries. They found high reversible capacity, ultralong cycling stability along with excellent rate capability [29]. To date, two-dimensional nanomaterials, particularly black phosphorous, have been used in supercapacitor application owing to the high specific surface area, and tunable and direct band gaps [30,31]. Moreover, Sun et al. reported sodium–sulfur batteries that were made up of TiO_2_ nanoparticles and showed great promise for practical applications in energy conversion and storage [32]. Moreover, pristine carbon-based materials such as activated carbon, graphite, and many more were used as electrode materials, though they have a high specific surface area and tunable pore size but poor mechanical and electrochemical properties compared to advanced carbon-based materials such as carbon nanotubes, graphene, and so on, which limit their uses as advanced functional materials. Additionally, they are being used as reinforcing materials in polymer matrix composite, in which they improved the properties of matrix a lot compared to pristine polymers. Secondly, polymer composite is easily synthesized and cost-effective along with eco-friendly materials and has been used as electrode materials for a long time; however, polymers materials are still suffered from poor mechanical strength. So, the combination of polymer with nanomaterials greatly enhances the performance of the resulting materials.

It has been reported that the combination of ZnS nanoparticles and carbon nanotubes decreased the bandgap of ZnS nanoparticles and increased the conductivity of the several composite folds by providing the passage for better electron transport [33]. Moreover, the size of the ZnS quantum dots can easily be tuned by the method of preparation, reaction parameters (processing time, temperature), nature, and amount of dopant used. Furthermore, the multiwalled carbon nanotubes (MWCNTs) have shown great interest as electrode materials due to their invaluable properties such as high stiffness, excellent electrical conductivity, along with a high aspect ratio; however, CNTs tend to agglomerate due to strong van der Waals forces. On another side, they are used in polymer matrices wherein they can significantly improve the general properties of CNT-based nanocomposites. So, to avail of its interesting properties, it is necessary to fully disperse the CNTs into the matrices; therefore, small-size nanoparticles, including quantum dots, can be used to improve the dispersion of CNTs. They incorporate between the tubes through interfacial communication and reduce the van der Waals interaction between them. Hence, the load transfer efficiency of the matrix can be improved by adding a small number of CNTs.

Nowadays, polymer matrix nanocomposite (PMC) plays a significant role in various applications by serving as an excellent platform for nanofillers. The proper integration of nanofillers into the porous network of polymers can improve the electron transfer mobility of the nanocomposite via enhanced interface interaction, arising at the nano level. So far, various conductive polymers such as polypyrrole [34], polyindole [35], and polythiophene [36] have been exploited in various applications. Among them, polyaniline serves as an excellent conductive polymer due to its easy synthesis routes, biocompatibility, and low cost. Thus, these properties make it a valuable polymer to choose in various combinations where all these mentioned qualities play a major role.

In this study, a nanocomposite that contains ZnS Qds (zero-dimensional), decorated on the surface of both MWCNT and AgNWs (one dimensional), in the polyaniline matrix was synthesized and characterized concerning the biofuel cell applications.

## 2. Experimental Section

### 2.1. Materials Used

Hydrated zinc nitrate (Zn (NO_3_)_2_·6H_2_O), sodium sulfide (Na_2_S), ethylene glycol (EG), *N*-methyl pyrrolidine (NMP), and buffers of phosphate (pH 5.0 and 7.0) were procured from Central Drug House, Pvt. Ltd. India, New Delhi, Delhi, India. Sodium dodecylbenzene sulfonate (SDBS) was purchased from Merck, Vikroli, Mumbai, India. Multiwalled carbon nanotubes (MWCNT), ferritin (10 mg mL^−1^ in 0.15 M NaCl), glucose oxidase (GOx) from Aspergillus niger, and glutaraldehyde were obtained from Sigma-Aldrich, Bengaluru, Karnataka, India. Aniline monomer was procured from Fisher Scientific, Powai, Mumbai, India. Other chemicals and reagents are used as received.

### 2.2. Instruments Used

The Fourier transform infrared spectroscopy (FTIR) was used to obtain the FTIR spectra by using Nicolet iS50 FT-IR instrument operating in the range of 4000–500 cm^−1^. X-ray diffraction (XRD) analyses were carried out by using the Rigaku Smart Lab X-ray diffractometer. Through these techniques, the functional groups and the crystallinity of the materials were determined. Additionally, scanning electron microscopy (SEM) (JSM, 6510 LV, JEOL, Tokyo, Japan) and transmission electron microscopy (TEM) (TEM 2100, JEOL, Tokyo, Japan) operated at 200 kV on a carbon-coated copper grid were used to examine the morphology and particle size of the synthesized nanocomposites. The SEM analyses were coupled with energy dispersive X-ray (EDX) spectroscopy and elemental mapping. Furthermore, electrochemical testing was performed using a three-electrode system, wherein modified GCE served as a working electrode, platinum wire as a counter, and Ag/AgCl (3 M KCl) as a reference electrode.

### 2.3. Green Synthesis of Zinc Sulfide Quantum Dots (ZnS Qds)

Zinc sulfide quantum dots (ZnS Qds) were synthesized as reported in the literature [37]. In a typical procedure, Zn (NO_3_)_2_·6H_2_O (1 M, 1.487 g) was mixed in 5 mL of double distilled water under sonication for 10 min to obtain a clear solution. To this solution, 10 mL of neem (Azadirachta indica) leaves extract was mixed dropwise with continuous stirring (30 min). Then, a homogenous solution (10 mL) of Na_2_S (2 M, 0.7808 g) was added dropwise to the above solution and agitated on a magnetic stirrer at ambient conditions (pH = 6.1) until the entire solution turned white. The whole mixture was refluxed for 1 h until the yellow yolk-colored solution was obtained. The yellow powder was obtained by centrifugation at 8000 rpms followed by drying in an oven overnight.

### 2.4. Preparation of MWCNT/ZnS/AgNWs/PANI Nanocomposite

The nanocomposite (MWCNT/ZnS/AgNWs/PANI) was prepared with slight modification, as reported by Inamuddin and Shakeel [38]. Briefly, 50 mg of MWCNT, 0.08 M aqueous SDBS solution, and 10 mL suspension of AgNWs were added in 50 mL (1 M HCl) solution. The AgNWs were prepared using the method reported by Gebeyehu et al. [39]. A 50 mg ZnS Qds as prepared above and 0.4 M aniline monomer in 50 mL of 1 M HCl solution was sonicated for 60 min. After that, the solutions of MWCNT, AgNWs, and ZnS Qds were mixed and sonicated for 10 min, followed by refluxing the mixture for 30 min at 60 °C. After cooling the solution to room temperature, an already sonicated solution of aniline was mixed into it and placed on magnetic stirring for 30 min. Next, ammonium persulphate (10.50 g, 0.46 M) prepared in 1 M HCl was added dropwise while keeping the resultant solution in an ice bath. The mixture was left on the ice bath to complete the polymerization reaction for 5–6 h under continuous magnetic stirring at 0–5 °C. The concentration ratio of molarity of APS to aniline and SDBS to aniline were 1.15:1 and 1:5, respectively. The blackish-green color mass was obtained after washing and drying in a vacuum oven at 60 °C for 24 h. The end-product was made ready for characterization after converting into a fine powder using a mortar pestle. The steps involved in the synthesis of the nanocomposite are shown in Figure 2a,b.

### 2.5. Preparation of MWCNT/ZnS/AgNWs/PANI Nanocomposite Suspension

Suspension of 20 mg of MWCNT/ZnS/AgNWs/PANI nanocomposite was prepared in 10 mL NMP solvent. Additionally, a UV-vis spectrophotometer was used for checking dispersion in the range between 300–700 nm.

### 2.6. Preparation of MWCNT/ZnS/AgNWs/PANI/Frt/GOx Bioanode

Firstly, a glassy carbon electrode (GCE) having a 3 mm diameter was cleaned gently by using alumina slurry on a velvet pad up to the mirror glass. GCE was sonicated followed by washing with both DDW and ethanol. After drying at room temperature, 6 µL suspension of the nanocomposite was cast on the bare GCE and left to dry for 4 h at room temperature. Further, 5 μL of Frt mediator followed by 6 μL of GOx solution were drop cast on the nanocomposite modified GCE, respectively. A 1.5 μL glutaraldehyde (2% aqueous solution) as cross-linker was applied to ensure the strong connection between enzymes, mediator, and electrode. Lastly, the fabricated electrode was air-dried and dipped into DDW for a while to remove un-immobilized bioactive material and kept in the refrigerator at 4 °C for further use.

## 3. Results and Discussion

### 3.1. FTIR Analysis

The FTIR spectrum of the synthesized nanocomposite was compared with the spectra of ZnS QDs, AgNWs, and MWCNT as displayed in Figure 3. The vibration frequency at 624 cm^−1^ reflects the Zn-S bond, whereas other vibration frequencies such as 1108, 1388, 1626, and 3404 cm^−1^ could be attributed due to the presence of biomolecules (Terpenoids and flavonoids) available in neem leaves extract used for its green synthesis Figure 3a [37]. Moreover, all the characteristics of possible vibrations such as 1024 and 1636 cm^−1^ are likely due to the C-N stretching and C=O group of PVP modified AgNWs as observed in the spectrum of AgNWs, Figure 3b [38]. The spectrum of MWCNT in Figure 3c displayed the vibrations at 1040, 1632, 2846, and 3404 cm^−1^, which strongly corresponds to the presence of CO group, stretching vibrations of carbon and hydrogen (C-H) and water molecules (-OH) that are adsorbed on the surface of CNTs, respectively. The spectrum of the synthesized nanocomposite in Figure 3d manifested all the vibrations as found in the individual spectrum. It can be explained by electrostatic interaction amongst the negatively charged MWCNTs and AgNWs with the positively charged ZnS Qds [40,41].

### 3.2. XRD Analysis

The X-ray diffraction (XRD) analyses of the synthesized MWCNT/ZnS/AgNWs/PANI nanocomposite, quantum dots, nanowires, and nanotubes were carried out. Figure 4a showed the diffraction peaks at 28.68°, 47.62°, and 56.42°. This pattern is in good agreement with the sphalerite ZnS phase [42]. The synthesized silver nanowires displayed all strong characteristic peaks at 38.3°, 44.46°, 64.82°, and 77.82° [39] as shown in Figure 4b. The diffraction patterns of MWCNTs showed peaks at 26.1° and 43.8°, as shown in Figure 4c. These peaks correspond to the diffraction pattern of the hexagonal graphitic structure. Figure 4d depicted almost all the patterns of individual components of the synthesized nanocomposite (MWCNT/ZnS/AgNWs/PANI).

### 3.3. Zeta Potential

The net electrostatic charge of the synthesized ZnS Qds, MWCNTs, and AgNWs was determined by a zeta seizer as given in Figure 5. The biologically synthesized ZnS Qds showed a high positive value of 15.5 mV Figure 5a, which reflects that the strong electrostatic repulsive forces stabilized the quantum dots and prevented their agglomeration. Moreover, the net electrostatic charges on MWCNTs and AgNWs Figure 5b,c were found to be −11.1 and −0.23 mV, respectively. The MWCNTs, AgNWs, and ZnS Qds show strong electrostatic interaction due to the presence of opposite charges on their surface. It can be concluded that this interaction leads to the successful synthesis of nanocomposite [43].

### 3.4. SEM Analysis

Scanning electron microscopy (SEM) was carried out to reveal the surface morphology of the synthesized nanocomposites at a magnification of 10,000×. Figure 6a shows that MWCNT and AgNWs are well scattered in the polyaniline matrix of the synthesized nanocomposite (MWCNT/ZnS/AgNWs/PANI). This is due to the Π-Π stacking between the aromatic ring electrons of PANI and the hexagonal pattern of MWCNT electrons. As a result, highly accessible electron communication occurs between them. Furthermore, AgNWs served as nanofillers that filled the available voids and thereby improving the interfacial interaction at the nanoscale through their dispersion. Consequently, improves the electrical communication among them. In addition, randomly distributed ZnS Qds also appear in the image, which reflects ZnS Qds are filling the intervening space and making a strong connection among them. Figure 6b,c illustrate mapping images that reflect all the possible elements of the nanocomposite in a confined portion. The EDX spectrum of the nanocomposite showed the weight % of the elements present in the nanocomposite, including C (60.47%), N (31.64%), S (4.63%), Zn (0.95%), and Ag (0.71%), as given in Figure 6d. Albeit, zinc, and silver showed in a minor amount in the spectrum; tiny weight % shows their presence in the sample.

### 3.5. TEM

The synthesis of ZnS Qds was confirmed using a transmission electron microscope (TEM). Figure 7a reveals the spherical morphology of quantum dots along with the average particle size of 11 nm. Figure 7b shows all the components present in the synthesized nanocomposite. It showed that the ZnS Qds are spread over the surfaces of MWCNTs and AgNWs in the PANI matrix. It is suggested that ZnS Qds has a good affinity to the rest of the components.

### 3.6. Electrochemical Studies

The electrochemical studies of the fabricated electrodes were evaluated by using cyclic voltammetry in PBS solution as a supporting electrolyte. The voltammograms were taken by varying the potential between +1 to −1 V. As can be seen in Figure 8, the curve (a) of bare GCE displayed no electrochemical response, whereas the curve (b) of ZnS Qds showed the appreciable electrochemical activity. This response is likely due to the increased surface area and high electrochemical activity of zinc sulfide quantum dots (ZnS Qds). Comparably, the voltammogram of PANI showed a higher electrochemical activity than ZnS Qds curve (c) owing to progressive oxidation (or reduction) of polymer’s backbone, which delivers a redox current density of 0.3 mAcm^−2^ and −0.32 mA cm^−2^ at 0.62 V and −0.36 V, respectively. The nanocomposite curve (d) revealed an even higher redox-active response than those of the aforementioned electrodes. The higher redox-active response is anticipated from the synergistic effect of components of nanocomposite (MWCNT/ZnS/AgNWs/PANI). The superior conductivity of MWCNT and AgNWs exhibit a considerable effective surface area along with the high catalytic activity that produced a high current density of 4 mA cm^−2^. In addition, the highest current density of 5 mA cm^−2^ was obtained after applying the Frt and GOx on the nanocomposite modified GCE. It could be due to the electrostatic and Π-Π interaction among the components of nanocomposite and originated from the immobilized enzyme. These interactions might have the thiol linkage between AgNWs and the sulfur group of ZnS Qds and the stacking between the Π system of PANI and MWCNT electrons. Consequently, these interactions might have contributed to the substantial enzyme loading by providing strong binding sites along with a high effective surface area; therefore, the high current density along with reversibility due to the redox-active nature implies that the proposed nanocomposite served as an efficient host for the binding of the enzyme.

Additionally, the modified bioanode MWCNT/ZnS/AgNWs/PANI/Frt/GOx was tested in the presence and absence of glucose to determine its biocatalytic performance, as shown in Figure 9. Noticeably, a lower current density of 5.6 mA cm^−2^ (Figure 9a) was found in the absence of glucose. In contrast, a relatively higher current density of 7.6 mA cm^−2^ (Figure 9b) with a well-defined redox peak at 0.1 V, and −0.32 V, due to the reduction and oxidation of glucose oxidase enzyme at the forward and backward scan, was obtained in the presence of 50 mM glucose. This increment in catalytic current confirms the catalysis of glucose into gluconolactone with the ejection of two electrons. Further high current density is supported by the adequate wiring of the enzyme on the nanocomposite, which is likely due to the large surface area of the highly conducting CNT, AgNWs, and zero-dimensional ZnS quantum dots.

The effect of scan rate on the bio-electrocatalytic activity of fabricated bioanode (MWCNT/ZnS/AgNWs/PANI/Frt/GOx) was tested by changing the sweeping potential (scan rate) from 20 to 100 mVs^−1^ as shown in Figure 10a. The scan rate curve reflects the linear increase in anodic and cathodic current density as the potential increases. From this outcome, it can be concluded that the electrocatalytic reactions occurring on MWCNT/ZnS/AgNWs/PANI/Frt/GOx modified electrodes were quasi-reversible surface-controlled reactions. In addition, the respective peak current versus scan rate calibration curve is plotted as depicted in Figure 10b, which provides a linear relationship to linear regression equations. Ipa = 0.00005x − 0.0006, Ipc = −0.00009x + 0.0008 and correlation coefficient of 0.992 and 0.993, respectively. Thus, these results lead to a conclusion that the fabricated bioanode reveals the speedy MET kinetics.

Further, the kinetics of the modified anode in terms of rate constant (ks) was evaluated to be 5.8 s^−1^ at 100 mV s^−1^ by using a prevalent Laviron Equation (1) [44]. A comparison of the current density and the rate constant of the given modified bioanode with the reported ones is shown in Table 1 [38,44,45,46,47,48,49,50,51].
(1)$$\log k_{s}=\alpha \log\left(1-\alpha \right)+\left(1-\alpha \right)\;\log\alpha -\log\left(\frac{RT}{nFv}\right)-\alpha \left(1-\alpha \right){\left(\frac{nF∆Ep}{2.3RT}\right)}^{\;}$$

Here, α (charge transfer coefficient), *R* (8.314 JK^−1^), *T* (temperature, 298 K), *F* (Faraday’s constant, 96,485 Cmol^−1^), *n* (number of electron transfer, 2), *ν* (scan rate, 100 mVs^−1^), and ∆*Ep* = Epa-Epc.

The familiar Brown–Anson model indicated Equation (2) [42] was used to calculate the concentration of the immobilized enzyme on the surface of MWCNT/ZnS/AgNWs/PANI/Frt/GOx bioanode, which was evaluated to be 3.8 × 10^−7^ mol cm^−2^.
(2)$$I_{p}=\frac{n^{2}F^{2}I^{*}AV}{4RT}$$
where *Ip* (oxidation current at anode at 100 mV s^−1^ scan rate), *I** = surface concentration of the bioanode to be calculated, *A* = GCE surface area (0.07 cm^2^), *R* (gas constants = 8.314 J·K^−1^), *T* (298 temperature in Kelvin), *F* (Faraday’s constant; 96,485 Cmol^−1^), and *n* = 2 (no of electrons released during oxidation).

### 3.7. EIS Study

The Nyquist plots of the given modified electrodes were obtained by employing the electrical impedance spectroscopy (EIS) as shown in Figure 11. After the modification of GCE with nanocomposite MWCNT/ZnS/AgNWs/PANI, the charge transfer resistance (Rct) drastically reduced to 416 Ω, which signifies that nanocomposite modified electrode exhibited excellent electronic conductivity that facilitated the electron transfer through 3-dimensional architecture. On the contrary, the enzyme immobilized electrode MWCNT/ZnS/AgNWs/PANI/Frt/GOx delivered an elevated Rct of 470 Ω. It is likely due to the steric hindrance of the enzyme to the redox probes in accessing the electrode surface, confirming the successful anchoring of enzymes on the surface of the electrode. The parameters R_s_, C_dl_, and Z_w_ represent solution resistance, double-layer capacitance, and Warburg impedance, respectively, of the Randle circuit, as shown in the inset of Figure 11.

### 3.8. LSV Analysis

The effect of the molar ratio of glucose on the biocatalytic activity of the manufactured bioanode MWCNT/ZnS/AgNWs/PANI/Frt/GOx was assessed by using the linear sweep voltammetry (LSV) technique. From Figure 12a, it can be observed that by varying the glucose concentration from 10 to 60 mM, the current density linearly increases until the concentration reaches 50 mM, and beyond this concentration, the saturation in current density was noticed. From this, it can be estimated that the first increase is likely due to the available unoccupied sites on the bioanode (biocatalyst), after which the enzyme is saturated, and the current density remains constant regardless of increasing concentration. After obtaining the steady-state current density at each specific concentration, the respective calibration plot was drawn as presented in Figure 12b. The optimized current density was obtained to be 7.6 mA cm^−2^ at 50 mM glucose in PBS 7.0 as a supporting electrolyte.

## 4. Conclusions

Keeping in mind the toxic hazards of chemicals on nature, an environmentally benign method was used for the synthesis of ZnS quantum dots. Herein, the neem leaves extract was successfully utilized for the preparation of nanodots. Thenceforth, ZnS Qds decorated on the surface of MWCNTs and AgNWs dispersed in the polyaniline matrix were successfully synthesized via in situ polymerization routes. The three-dimensional network of the proposed nanocomposite assisted the electron tunneling by forming a continuous network accompanying various interactions between the fillers and polymer matrix. Further, it is worth noting that the sulfur atoms of the nanodots form a thiol linkage with the AgNWs, which prevents nanowires from oxidation by forming a protective covering over them. Consequently, such kinds of interactions improved the loading of the enzyme that in turn increasing the catalysis of the enzyme. In short, the proposed bioanode has a simple, cost-effective, time-feasible synthesis so that it can be used as the anode in glucose biofuel cell (GBFC) for electricity generation from glucose; therefore, the proposed bioanode could be feasible for commercialization and in practical applications not only in biofuel cells applications but also in the detection of glucose in biomedical applications.

## Figures and Tables

**Figure 1 polymers-14-01321-f001:**
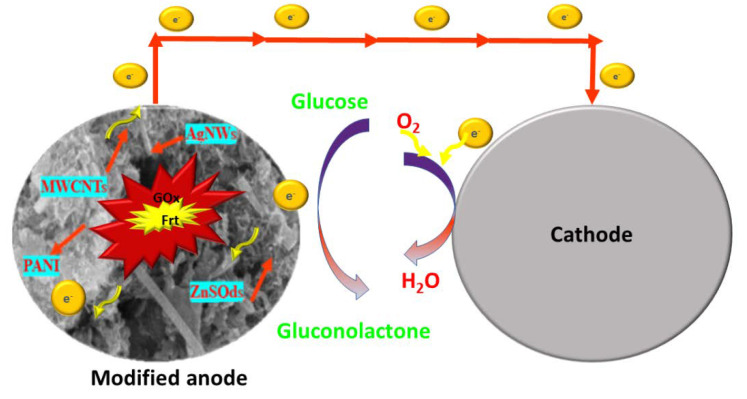
Illustrates the modified bioanode, depicts the shuttling of electrons via a biochemical pathway.

**Figure 2 polymers-14-01321-f002:**
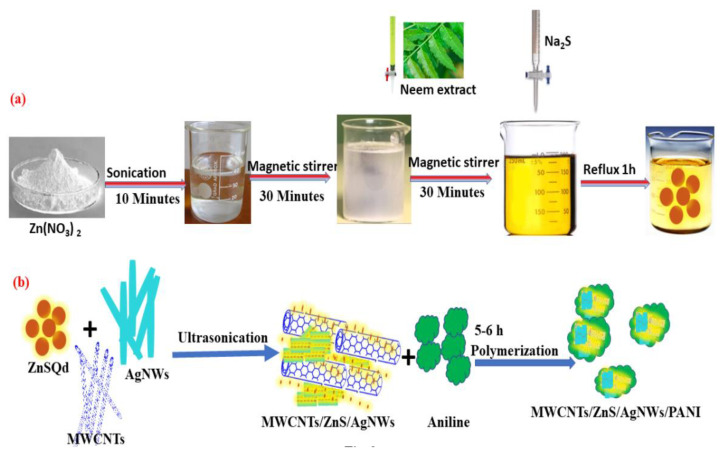
Schematic representation of step-by-step preparation of: (**a**) zinc sulphide quantum dots (ZnS Qds); (**b**) (MWCNT/ZnS/AgNWs/PANI) nanocomposite.

**Figure 3 polymers-14-01321-f003:**
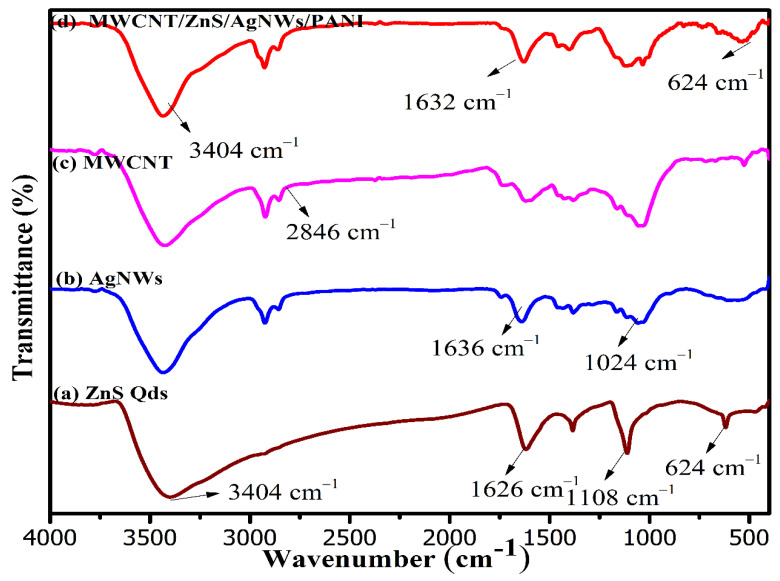
FTIR spectra of: (**a**) ZnS Qds; (**b**) AgNWs; (**c**) MWCNTs; (**d**) (MWCNT/ZnS/AgNWs/PANI) nanocomposite.

**Figure 4 polymers-14-01321-f004:**
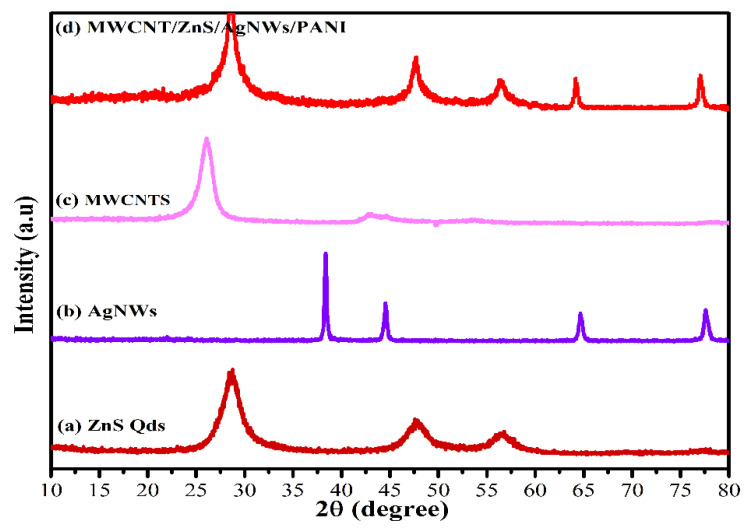
XRD pattern of: (**a**) ZnS Qds; (**b**) AgNWs; (**c**) MWCNTs; (**d**) (MWCNT/ZnS/AgNWs/PANI) nanocomposite.

**Figure 5 polymers-14-01321-f005:**
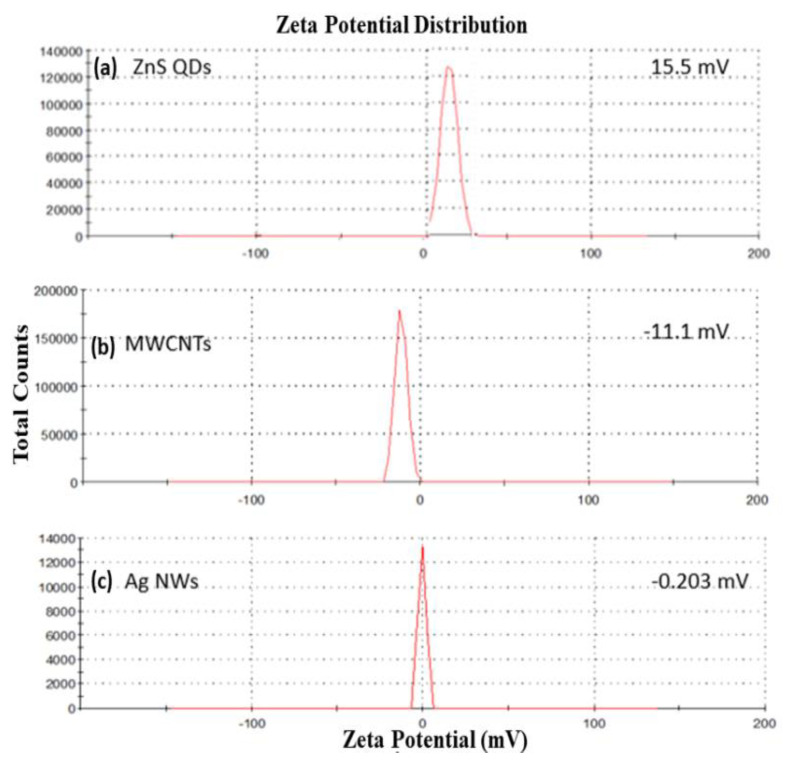
Zeta potential of: (**a**–**c**).

**Figure 6 polymers-14-01321-f006:**
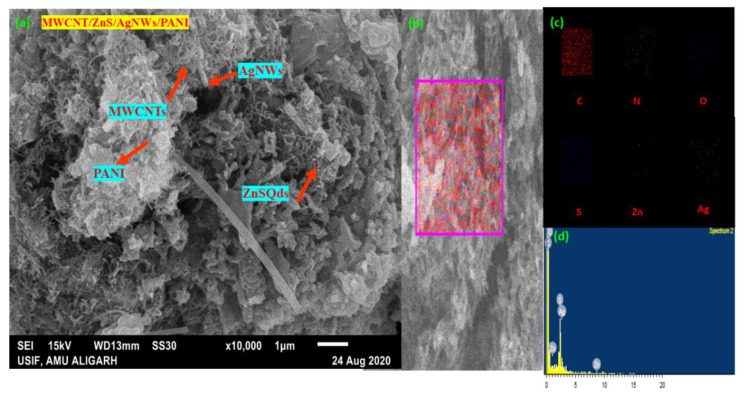
SEM images of: (**a**) (MWCNT/ZnS/AgNWs/PANI) nanocomposite; (**b**,**c**) mapping (**d**) EDX of nanocomposite.

**Figure 7 polymers-14-01321-f007:**
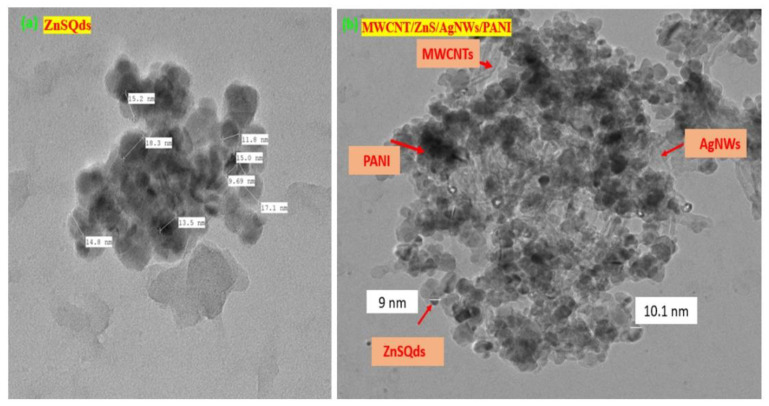
TEM images of: (**a**) ZnS Qds; (**b**) (MWCNT/ZnS/AgNWs/PANI) nanocomposite.

**Figure 8 polymers-14-01321-f008:**
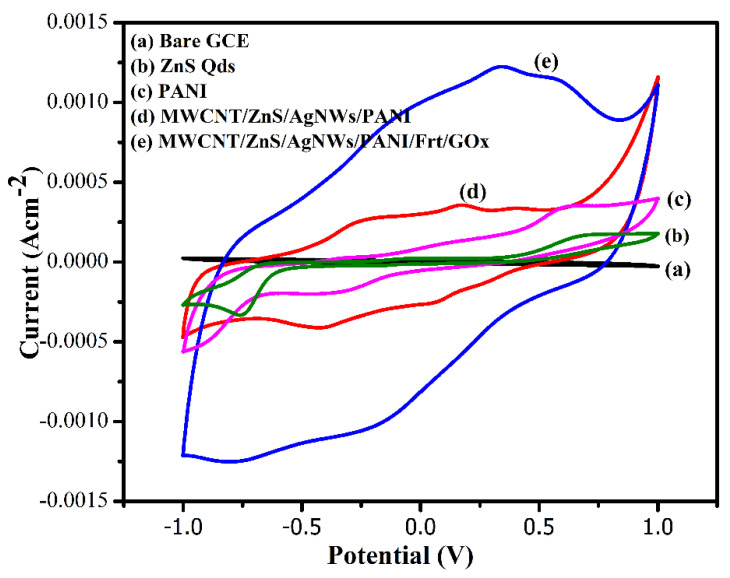
Cyclic voltammograms of: (**a**) bare GCE; (**b**) GCE/ZnS Qds; (**c**) GCE/PANI; (**d**) GCE/MWCNT/ZnS/AgNWs/PANI; (**e**) GCE/(MWCNT/ZnS/AgNWs/PANI/Frt/GOx in PBS pH 7.0 as supporting electrolyte at ambient conditions.

**Figure 9 polymers-14-01321-f009:**
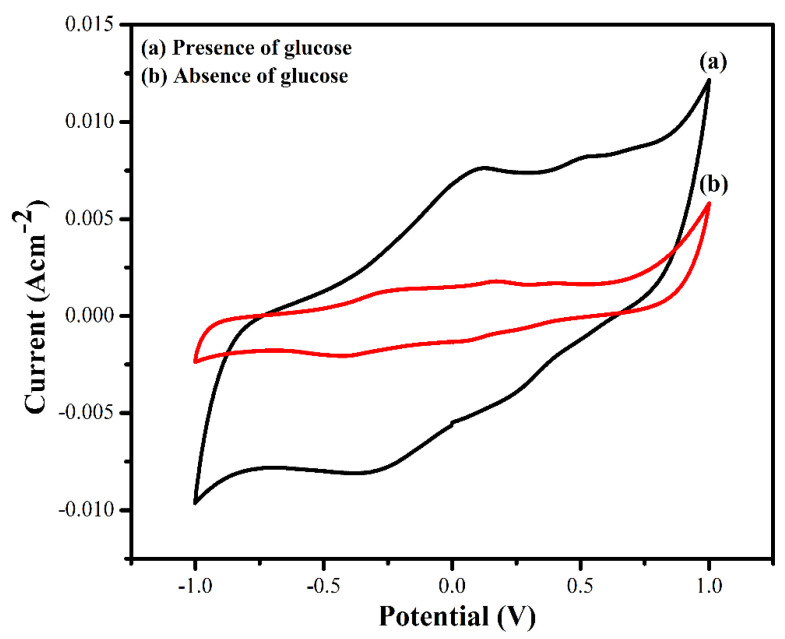
Cyclic voltammograms of GCE/(MWCNT/ZnS/AgNWs/PANI/Frt/GOx: (**a**) in presence of 50 mM glucose concentration; (**b**) absence of glucose.

**Figure 10 polymers-14-01321-f010:**
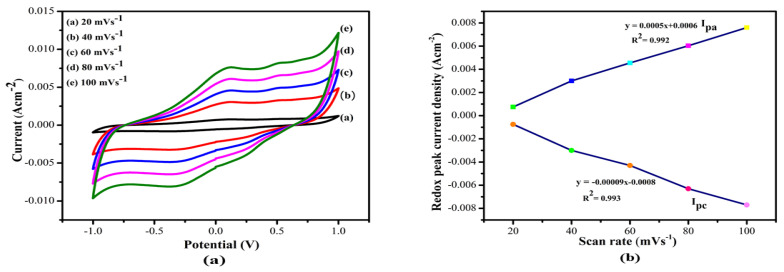
(**a**) Cyclic voltammograms of GCE/(MWCNT/ZnS/AgNWs/PANI/Frt/GOx at different scan rates ranging from 20–100 mV s^−1^ in the presence of 50 mM glucose. (**b**) Redox peaks calibration curve of the corresponding bioanode with the expansion in scan rate (20–100 mV s^−1^).

**Figure 11 polymers-14-01321-f011:**
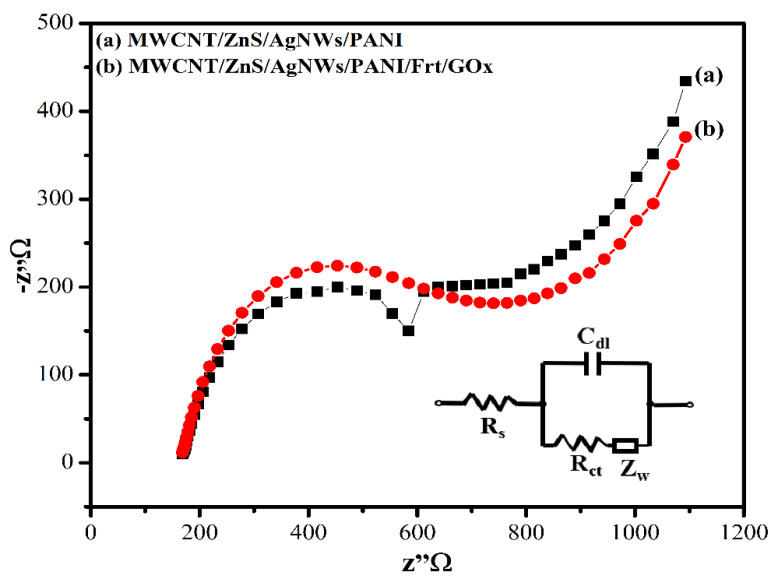
Nyquist plot of (**a**) GCE/(MWCNT/ZnS/AgNWs/PANI (**b**) GCE/(MWCNT/ZnS/AgNWs/ PANI/Frt/GOx.

**Figure 12 polymers-14-01321-f012:**
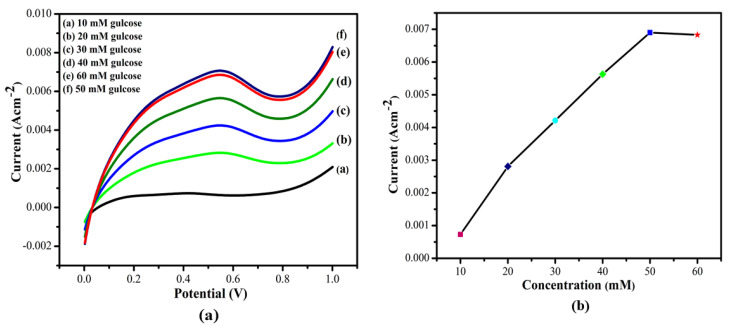
(**a**) Linear sweep voltammogram of GCE/(MWCNT/ZnS/AgNWs/PANI/Frt/GOx by increasing glucose range (10–60 mM) in PBS pH. 7.0 at ambient temperature. (**b**) The calibration curve of current densities observed in the glucose concentration ranged from 10–60 mM.

**Table 1 polymers-14-01321-t001:** Shows the current densities and rate constants (**k_s_**).

S.No.	Fabricated Bioanodes	Current Density (mA cm^−2^)	k_s_ (s^−1^)	Reference
1.	ZnO/PIn-MWCNTs/Frt/GOx	4.9	4.28	[38]
2.	Kraton/MWCNTs/Frt/GOx	1.14	1.83	[44]
3.	(f-SWCNTs@Ppy@NiMoSe_2_/Frt/GOx)	9.01	15.6	[45]
4.	GCE/MnO_2_-G/PTA/Frt/GOx	3.68	1.82	[46]
5.	Ppy-Ag-GO/Frt/GOx	5.7	1.59	[47]
6.	PPy/Au/CNT@Fe_3_O_4_/FRT/GOD	6.01	3.74	[48]
7.	PTH-TiO_2_/Frt/GOx	7.8	18.32	[49]
8.	Ti_3_C_2_ MXene	2.1	-	[50]
9.	GCE/Os(bpy)2PVI/GOx bioanode	5.30	-	[51]
10.	MWCNT/ZnS/AgNWs/PANI/Frt/GOx	7.6	5.8	Present work

## Data Availability

Data will be made available on request.

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
