# Peer review of "ZnS Quantum Dots Decorated on One-Dimensional Scaffold of MWCNT/PANI Conducting Nanocomposite as an Anode for Enzymatic Biofuel Cell"

_polymers, 2022, doi:10.3390/polym14071321_

Round 1

Reviewer 1 Report

In this manuscript, a new nanocomposite for the wiring of the enzyme and its mediator used as a bioanode in the EBFC is reported. ZnS quantum dots were synthesized first and the nanocomposite was synthesized by in-situ polymerization of aniline monomer. The synthesized MWCNT/ZnS/AgNWs/PANI was characterized by various analytical techniques. It has been proved that the proposed bioanode could be feasible for commercialization and in practical applications not only in BFC applications

I consider the content of this manuscript will definitely meet the reading interests of the readers of the Polymers journal. Therefore, I suggest giving a minor revision and the authors need to clarify some issues or supply some more data to enrich the content.

  1. Abstract and Introduction

  • The length of the abstract is a little longer, which exceeds the requirements of the journal. Hence, the abstract should be further reduced a bit. ‘Abstract: The abstract should be a total of about 200 words maximum’ (https://www.mdpi.com/journal/polymers/instructions).

  • For the Keywords, ‘polyaniline/PANI’, ‘in-situ polymerization’, ‘nanofiller’, and ‘electrochemical behaviour’ should also be added to attract a broader readership and highlight the significance of this work.

  • Please pay attention to grammar and spelling problems, especially the missing or redundant definite articles. I suggest double-checking the whole manuscript. I will point out several examples, but unfortunately, I cannot point out all of them. For example:

The title should be ‘conducting’, not ‘MWCNT/PANI conduting’;

Abstract part, ‘... the designed bioanode underwent into the electrochemical assessment ...’;

Page 4, ‘2. Materials and Methods, 2. Experimental Section’ should not appear twice;

Page 6, ‘The X-ray diffraction (XRD) analyses of ... and nanotubes were carried out’;

Page 13, ‘Consequently, such kinds of interactions improved the loading of the enzyme that in turn increasing the catalysis of the enzyme’ and so on.

  • Page 1, the introduction of EBFC is quite sufficient. However, as an electrochemical generator, how about the comparison with other electrochemical systems, in terms of cost, lifetime, maximum current density/power density/energy density, operating potential, etc., such as PEMFCs and flow batteries [Materials today 32 (2020): 178-203; Journal of Power Sources 493 (2021): 229445]? This should be briefly discussed as well since EBFC is not that well-known and widely used.

    In addition, for most electrochemical generators, a membrane/separator is typically used as a barrier layer between the cathode and anode [Electrochimica Acta 378 (2021): 138133]. However, for EBFC, the lack of need for a membrane simplifies the cell design to be small and compact, given that hydrogenase does not react with oxygen (an inhibitor) and the cathode enzymes (typically laccase) does not react with the fuel. This is another unique and significant feature of EBFC that should be explained to the readers.

  • Page 3, until the end of the Introduction part and after introducing so many advantages for different single components, I still find nothing about what is the commonly used electrode materials in EBFC. That is a missing part in the Introduction section.

    In addition, compared with the noble metal catalysts used in conventional fuel cells, for EBFC these enzymes can be obtained from lower-cost renewable raw materials. But the use of AgNWs (one dimensional) is also quite expensive, so I want to ask, is the practicability, scale-up potential and cost performance of the obtained composite bioelectrode really so high?

  1. Materials and Methods

Page 4, ‘A 50 mg ZnS Qds as prepared above and 0.4 M aniline monomer in 50 mL of 1 M HCl solution were sonicated for 60 minutes.’ 

Since polyaniline is prepared by doping acid, the different morphology and growth modes of polyaniline may also affect the final properties of the composites. Did the author consider preparing the polyaniline nanotube structure? It is widely reported that polyaniline nanotubes have better electrochemical performance due to the high effective surface area as well [Advanced Functional Materials 13.10 (2003): 815-820; Advanced materials 24.9 (2012): 1176-1181]. Is the preparation route of polyaniline selected by the author the optimal solution?

  1. Results and discussion

  • Page 7 and 8, ‘ 6 (a) showed that MWCNT and AgNWs are well scattered in the polyaniline matrix of the synthesized nanocomposite (MWCNT/ZnS/AgNWs/PANI).

So what is the exact or concrete morphology for the PANI matrix? Is it nanoparticles, nanosheets or nanorods? I can't be sure of this from SEM and TEM morphology.

  • Page 8, ‘The EDX spectrum of the nanocomposite showed the weight % of the elements present in nanocomposite including C (60.47%), N (31.64%), S (4.63%), Zn (0.95%) and Ag (0.71%) as given in Fig. 6 (d).’

What is the conclusion that can be drawn from the EDX spectrum result? Zinc and silver are both in tiny amounts, but is the result consistent with the theoretical value? There need to be some more explanations of the EDX results.

  • Page 9, ‘These interactions might have the thiol linkage between AgNWs and the sulphur group of ZnS Qds, hydrogen bonding between the oxygen of MWCNTs and the N-H of PANI...

What is the oxygen of MWCNTs? It needs to be explained better.

  • Page 11, ‘Table 1. Shows the current densities and rate constants (ks) values of the fabricated bioanode along with the reported ones.

It can be a sentence, but it is not suitable for the caption of Table 1 starts with Shows the current densities...’.

  • In the electrochemical characterization part, I do not deny that CV Curve, EIS Study and LSV analysis all demonstrate the electrochemical properties of the prepared composite electrode materials. However, I still consider there are two important deficiencies in the experimental design in this part, which should be supplemented:
  • Why is electrochemical characterization only compared with the electrode prepared by the authors, rather than with the most widely used electrode (commercial electrode or standard electrode) in EBFC? I consider the practicability and general applicability of only comparing with the different electrodes prepared by the authors are very limited.
  • Since the prepared electrode is designed for the EBFC applications, the most direct and effective testing method is to apply the electrodes prepared in this paper to EBFCs for in-situ cell test and compare the results with the performance of the current EBFC system equipped with traditional electrodes. This is the most direct evidence, which proves that the electrode materials obtained in this paper have advantages in the application of EBFC. However, this part is still missing, and such an issue should be addressed if possible.

Author Response

Reviewer #: "This manuscript presented new composite electrode material for enzyme
biofuel cell. Although this work might have a great interest from the
audience, there are certain improvements required:

Comment 1:  

Figures quality. First of all, all your images are of different
formats and styles. Sometimes fonts are awkward and stretched. Please
re-work Figs 8-9-10 and you need to polish some other Figs to make them
look better

Response 1:      

As per the reviewer’s suggestion quality of the figures have been improved.

Comment 2:           

FT-IR graph (Fig3) All spectra are without assignments for the peaks.
Please make the graph of better look and assign all-important peaks.

Response 2:      

As per the reviewer’s suggestion the important peaks have been marked in the graph.

Comment 3   

Nyquist plot.  Please provide the equivalent circuit and make the
graph better quality. 

Response 3:      

As per the reviewer’s suggestion the equivalent circuit have been given in the inset of the figure and resolution has also improved.

Comment 4:   

The reason for the redox peaks in the cyclic voltammograms (Figures 6,7) should be explained.

Response 4

As per the reviewer’s suggestion the reason of the redox peak in the voltammograms have been explained.

Comment 5:   

Please explain the decision for the choice of the Polymers journal.
Why not catalysts or nanomaterials?"

Response 5

The authors have chosen Polymer journal because firstly this is one of the esteem peer-review journals and we have synthesized conducting polymer nanocomposite.

We have seen the Special Issue which is directly related to our work so we have submitted the paper in the special issue.

Conducting Polymer Nanocomposites and Their Potential Applications

edited by 

Mohammed Muzibur Rahman

submission deadline 20 Mar 2022 | 5 articles | Viewed by 2453

Keywords: conducting polymers; nanofibers; composite materials; hybrid materials; chemical sensors; co-polymers; bio-sensors; electrosping; rubber composites; bio-degradation; fibrous materials; carbon materials; polymer blends; polymerization; biopolymers; bioactive polymers; drug delivery; hydrogels; fabrication; sol-gel; cross-linking; bioplastic; polymer nanomaterials; celluloses

(This special issue belongs to the Section Polymer Applications)

Basically, we have reinforced the polymer by adding different nanofillers and studied electrochemical study of the same. Therefore, this journal is suitable for this study and we will glad to see our article in this esteem journal.

Reviewer 2 Report

This manuscript presented new composite electrode material for enzyme biofuel cell. Although this work might have a great interest from the audience, there are certain improvements required:

1) Figures quality. First of all, all your images are of different formats and styles. Sometimes fonts are awkward and stretched. Please re-work Figs 8-9-10 and you need to polish some other Figs to make them look better

2) FT-IR graph (Fig3) All spectra are without assignments for the peaks. Please make the graph of better look and assign all-important peaks.

3) Nyquist plot.  Please provide the equivalent circuit and make the graph better quality. 

4) Please explain the decision for the choice of the Polymers journal. Why not catalysts or nanomaterials?

Author Response

(The authors gave the same response as above.)

Round 2

Reviewer 2 Report

Authors have made sufficient improvements but the quality of Fig6 is still poor. I cannot read the digits. Please rework Fig6.